# Using historical temperature to constrain the climate sensitivity, the transient climate response, and aerosol-induced cooling

Olaf Morgenstern[1]

[1]National Institute of Water and Atmospheric Research (NIWA), Wellington, New Zealand

**Correspondence:** Olaf Morgenstern (olaf.morgenstern@gmail.com)

**Abstract.** The most recent generation of climate models that has informed the 6[th] Assessment Report (AR6) of IPCC is characterized by the presence of several models with larger equilibrium climate sensitivities (ECSs) and transient climate responses (TCRs) than exhibited by the previous generation. Partly as a result, AR6 did not use any direct quantifications of ECSs and TCRs based on 4×CO2 and 1pctCO2 simulations and relied on other evidence when assessing the Earth's actual ECS and TCR. Here I use historical observed global-mean temperature and simulations produced under the Detection and Attribution Model Intercomparison Project to constrain the ECS, TCR, and historical aerosol-related cooling. I introduce additivity criteria that disqualify 8 of the participating 16 models from consideration in multi-model averaging calculations. Based on the remaining 8 models I obtain an average adjusted ECS of $3.5\pm0.4$ K and a TCR of $1.8\pm0.3$ K (both at 68% confidence). Both are consistent with the AR6 estimates but with substantially reduced uncertainties. Furthermore, importantly I find that the optimal cooling due to short-lived climate forcers consistent with the observed temperature record should on average be about $47\pm39\%$ of what these models simulate in their aerosol-only simulations, yielding a multi-model-mean, global-, and annual-mean cooling due to near-term climate forcers for 2000-2014, relative to 1850-1899, of $0.24\pm0.11$ K (at 68% confidence). This is consistent with but at the lower end of the very likely uncertainty range of IPCC. There is a correlation between the models' ECSs and their aerosol-related cooling, whereby large-ECS models tend to be associated also with large aerosol-related cooling. The results imply that a reduction of the aerosol-related cooling, along with a more moderate adjustment of the greenhouse-gas related warming, for most models would bring the historical global mean temperature simulated by these models into better agreement with observations.

## 1 Introduction

The equilibrium climate sensitivity (ECS) is a well-established (Arrhenius, 1896) yet, despite progress, poorly constrained property of the climate system (Knutti et al., 2017; Forster et al., 2021; Smith and Forster, 2021). For a hypothetical doubling of the atmospheric $CO_2$ content above preindustrial levels, it states the associated surface temperature increase at equilibrium. Similarly, the transient climate response (TCR) measures the warming simulated in simulations with $CO_2$ increasing at 1% per annum (p.a.) above its preindustrial abundance at the time of $CO_2$ doubling. For both quantities, disagreement amongst climate models, particularly in the most recent generation (Meehl et al., 2020), persisting despite ever improving model physics and resolution, is an impediment to narrowing their associated long-standing uncertainties. The large spreads in ECSs and

TCRs characterizing the present generation of climate models are contributing to some substantial inter-model spread in simulated end-of-century warming in future-scenario simulations (Lee et al., 2021). It is therefore desirable to reduce these model disagreements to more confidently project future climate under any climate scenario.

A standard optimal detection framework assumes that (a) changes in an observed or simulated quantity such as temperature are the sum of changes driven by individual climate forcers (referred to as "additivity") in the presence of climatological noise, and (b) model imperfections can be captured by introducing scaling or correction factors to those model responses to external climate forcers. This amounts, in a regression analysis, to optimally reproducing the observed variations and thereby constraining the TCR and ECS (Schurer et al., 2018, and references therein). The presence of climatological noise means that attribution benefits from using as many models and simulations as possible. Approaches of this kind have been used in many studies using CMIP5 and earlier model generations (e.g. Hasselmann, 1993; Hegerl et al., 1997; Allen and Tett, 1999; Schurer et al., 2018) and again for the most recent generation, CMIP6 (Gillett et al., 2021). When separating the GHG and aerosol influences, the main problem is that GHGs and aerosols cause warming and cooling effects on climate that are similar in terms of overall temporal developments and therefore can be difficult to distinguish statistically in observational or simulated records of temperature (see below). Furthermore, especially before CMIP6 single-model ensembles used to be small (the largest single-model ensemble used by Schurer et al. (2018) had 6 ensemble members). These two problems, along with general model disagreements on the sizes of these effects, combine to yield substantial uncertainties characterizing successive quantifications of TCR, ECS, and aerosol-induced cooling, such as for CMIP6 1.2 to 1.9 K for the TCR and $-0.7$ to $-0.1$K for the aerosol-induced cooling (Gillett et al., 2021). Both uncertainties contribute to uncertain projections of future global warming, e.g. 2.1 to 3.5 K of warming in 2081-2100, relative to 1850-1900, in the middle-of-the-road Shared Socioeconomic Pathway (SSP) 245 (Lee et al., 2021).

In this work I explore what "historical", all-forcings experiments and single-forcing experiments conducted for the Detection and Attribution Model Intercomparison Project (DAMIP, Gillett et al., 2016) imply for the ECS, TCR, and the aerosol-driven cooling which partially offsets global warming.

Decreases in future aerosol loading are thought to contribute to projected warming (Andreae et al., 2005), but the size of this effect is highly uncertain (Forster et al., 2021; Watson-Parris and Smith, 2022). In individual CMIP6 models mismatches between observed and simulated "historical" global-mean surface temperatures have been associated with a misrepresentation of the aerosol-induced cooling (Andrews et al., 2020; Smith and Forster, 2021; Golaz et al., 2022), such that despite the increases in the mean ECS and TCR characterizing the 6[th] Coupled Model Intercomparison Project (CMIP6) ensemble of models, relative to CMIP5, the simulated historical warming in CMIP6 is actually smaller than in CMIP5 (Flynn and Mauritsen, 2020; Smith and Forster, 2021; Flynn et al., 2023). Smith and Forster (2021) assess that differences between CMIP5 and CMIP6 "historical" simulations are due to an increased ensemble-mean climate sensitivity in CMIP6 versus CMIP5, compensated by a marginally increased aerosol radiative forcing and associated cooling. This causes lower temperatures during 1960-1990 and a larger post-1990 warming trend in the CMIP6 models (Flynn et al., 2023).

The CMIP6 generation of climate model simulations differs from previous generations in some important respects:

– The CMIP6 ensemble now contains several single-model "large" ensembles (i.e. consisting of at least 10 ensemble members, table 1), including for some single-forcing experiments.

– Studies targeting CMIP5 generally only used single-forcing simulations for the GHG and natural influences, inferring the influence of aerosols from these and the "historical", all-forcings simulations (Schurer et al., 2018). Here I will exploit single-forcing simulations explicitly targeting anthropogenic aerosols (as have Gillett et al., 2021). Using these "hist-aer"
simulations means I will explicitly test for and not simply assume additivity, as has been underlying these earlier studies.

– Since the 1990s, aerosol forcing has been on a declining trend (Szopa et al., 2021; Hodnebrog et al., 2024). This trend reversal means that the aerosol influence is becoming better distinguishable from the dominant influence of the ever-increasing GHG-induced warming. The additional years in the CMIP6 DAMIP simulations since this turnaround might critically improve the detectability of the aerosol influence.

A fundamental problem remains that nature has produced only one realization which might deviate in unknown ways from what the average of a large ensemble of hypothetical such realizations would indicate. This problem probably requires a probabilistic approach to tackle, which would be a further development on the method laid out below. I will return to this problem in section 4.

Below I develop a regression approach to evaluate, for all models participating in DAMIP, the fidelity of the simulation of
both GHG-driven surface warming and aerosol-related cooling in the CMIP6 ensemble.

## 2   Data and Method

### 2.1   DAMIP models and experiments

Simulations produced for DAMIP, "historical" simulations (Eyring et al., 2016) extended to 2020 using the (SSP245 simulations (Gidden et al., 2019), and published ECS and TCR values that are based on 4×CO2 and 1pctCO2 simulations (Eyring
et al., 2016) form the basis of this analysis. I use all 16 models for which "hist-GHG", "hist-aer", and "hist-nat" simulations are available (table 1). The hist-GHG, hist-aer, and hist-nat experiments are identical to the "historical", coupled all-forcings simulations except that forcings other than the GHGs, aerosols or their precursors, and natural (solar, volcanic) influences, respectively, are held at their 1850 values (Gillett et al., 2016).

The models that have completed these simulations span large ranges of ECSs (between 2.5 and 5.6 K) and TCRs (between
1.5 and 2.7 K). For 12 models the simulation period for DAMIP simulations is 1850-2020, but four models (CESM2, E3SM-2-0, GISS-E2-1-G, MPI-ESM1-2-LR) end their DAMIP simulations in 2014.

Furthermore as an observational reference I use the HadCRUT5 global-mean temperature climatology (Morice et al., 2021). The choice of this reference is not crucial; other reference datasets yield very similar results to those shown here. I form ensemble-, global-, and annual-mean temperature timeseries from the available simulations.
HadCRUT5 is an amalgamate of sea-surface temperatures and near-surface air temperatures over land and sea ice (Morice et al., 2021). For a rigorous comparison with the temperature fields provided by the CMIP6 models, I thus form, individually

for every model, similar amalgamates of "surface temperature" (ts) and "surface air temperature" (tas) using

$$T = \overline{(1 - f - s)T_s + (f + s)T_{sa}} \tag{1}$$

where $T$ is the global-, annual-, and ensemble-mean temperature used in the below analysis, $T_s$ is the monthly-mean, latitude-longitude resolved surface temperature, $T_{sa}$ is the monthly-mean surface-air temperature, $f$ is the land fraction at every grid-point, $s$ the monthly-mean sea ice fraction, and the overline marks global and annual averaging. For sea ice I use sea ice concentration on the ocean grid (siconc), regridded to the atmosphere grid using nearest-neighbour interpolation. Only for the GISS-E2-1-G model I use sea ice concentration on the atmosphere grid (siconca) because of unavailability of the siconc variable.

Furthermore, to reduce the influence of interannual variations, I apply a 15-year boxcar filter to the hist-aer and hist-GHG ensemble means, but not to the hist-nat, historical, and HadCRUT5 datasets. This choice is based on the understanding that interannual variations in hist-aer and hist-GHG reflect random climate variability which I do not expect to correlate with the climate variability in the "historical" ensemble or in HadCRUT5. Volcanic eruptions produce forced variations on the annual timescale in the hist-nat and "historical" ensembles and in HadCRUT5 which I want to preserve. Hence the asymmetric application of the filter.

Figure 1 summarizes the behaviour of surface/surface-air temperature in the 16 models over the historical period. Panel (a) indicates that as there is an approximate proportionality between the simulated warming attributable to GHGs (as taken from the hist-GHG ensembles) and the models' tabulated ECS and TCR values. Furthermore, panel (b) indicates that with some notable exceptions, the models are mostly additive in the sense that the sum of the warmings simulated in the three DAMIP experiments is mostly quite similar to the warming simulated in the models' historical ensembles.

## 2.2 Regression models

Using a linear regression approach, I derive rescaling factors $\alpha_1$, $\beta_1$, $\gamma_1$, $\alpha_2$, $\beta_2$, and $\gamma_2$ for the temperature responses to GHG, aerosol, and natural forcings such that the resultant sums of the rescaled simulated temperature anomalies minimize the root-mean-squared deviations $\epsilon_1$ and $\epsilon_2$ versus the ensemble-, global-, and annual-mean "historical" temperature anomaly $T_{hist}$ and the HadCRUT5 temperature anomaly record $T_{obs}$, respectively, over the period 1850-2020 or 2014 (171 or 165 years), respectively:

$$T_{obs} = \alpha_1 T_{hGHG} + \beta_1 T_{haer} + \gamma_1 T_{hnat} + \delta_1 + \epsilon_1 \text{ and} \tag{2}$$

$$T_{hist} = \alpha_2 T_{hGHG} + \beta_2 T_{haer} + \gamma_2 T_{hnat} + \delta_2 + \epsilon_2. \tag{3}$$

$T_{hGHG}$, $T_{haer}$, and $T_{hnat}$ are all normalized relative to their 1850-1899 average. $\delta_1$ and $\delta_2$ are intercepts that account for uncertainties (due to natural variability and other factors) in this normalization process, i.e. the other regression coefficients are insensitive to the normalization. $\epsilon_1$ and $\epsilon_2$ are the regression residual timeseries. This approach is similar to Gillett et al. (2021).

For a model which is perfectly additive in the anthropogenic GHGs, anthropogenic aerosol, and natural forcings, in the absence of climatological noise, and if the regression model was complete, the regression coefficients $\alpha_2$, $\beta_2$, and $\gamma_2$ would be 1. However, omitted here is the influence of ozone. Since 1900, the temperature changes due to ozone have been around $-30\%$ of that of the aerosols in the IPCC's best estimate (figure 7.8 of Forster et al., 2021). Therefore, the terms $\beta_1 T_{haer}$ and $\beta_2 T_{haer}$ are going to capture the contributions of both aerosol and ozone changes (i.e. the "near-term climate forcers", NTCFs) to the evolutions of $T_{obs}$ and $T_{hist}$. Because of the offsetting role of ozone, I thus expect $\beta_2 < 1$ for the perfect model, whereas for the GHG influence I expect $\alpha_2 \approx 1$.

With this understanding, I introduce thresholds

$$0.8 \leq \alpha_2 \leq 1.2 \tag{4}$$

and

$$0.6 \leq \beta_2 \leq 1.1 \tag{5}$$

to identify and define models that satisfy additivity, and only use only those models satisfying both criteria in an "emergent constraint" approach, to calculate best estimates and uncertainty ranges for the ECS, TCR, and aerosol-induced cooling (see below). This is the main difference w.r.t. Gillett et al. (2021) who use all models without considering their additivity properties. Essentially these two conditions remove models from multi-model emergent-constraint calculations if their regression coefficients versus $T_{hist}$ deviate substantially from expectations. I will discuss the role of ozone separately below.

I note that there is a considerable joint uncertainty resulting from substantial anticorrelations between $T_{hGHG}$ and $T_{haer}$, but much smaller correlations (and practically no joint uncertainty) between $T_{hGHG}$ and $T_{nat}$, and between $T_{haer}$ and $T_{nat}$. This allows me to simplify the analysis and focus in the following only on the GHG and aerosol influences. To study their joint uncertainty, I calculate, as a function of $\alpha$ and $\beta$ and time, the regression error timeseries

$$r_{obs}(\alpha, \beta) = T_{obs} - \alpha T_{GHG} - \beta T_{haer} - \gamma_1 T_{hnat} - \delta_1 \tag{6}$$

I thus define an error function

$$E_{obs}(\alpha, \beta) \equiv \sqrt{\frac{\overline{r_{obs}^2(\alpha, \beta)}}{\overline{\epsilon_1^2}}}, \tag{7}$$

where the overbar denotes the 171 and 165-year means over 1850-2020 and 1850-2014, respectively.

I define that a regression fit differs from the optimal fit of equation 2 if $E_{obs} > E_{max}$, where $E_{max}$ is a value of the error function, to be determined below, where the fits associated with such an RMS residual differ significantly from the optimal fit.

Substituting $Q_{obs}(\alpha, \beta) = \overline{r_{obs}^2(\alpha, \beta)}$, with $\Delta\alpha = \alpha - \alpha_1$ and $\Delta\beta = \beta - \beta_1$, I express $Q_{obs}$ as a quadratic form expanded around the minimum:

$$Q_{obs}(\alpha, \beta) = \overline{\epsilon_1^2} + \frac{1}{2}(\Delta\alpha, \Delta\beta)\mathbf{M}\begin{pmatrix}\Delta\alpha \\ \Delta\beta\end{pmatrix} \tag{8}$$

where terms linear in $(\Delta\alpha, \Delta\beta)$ are 0 because the expansion is around the minimum of $Q_{obs}$, and

$$\mathbf{M} = \begin{pmatrix} \frac{\partial^2}{\partial\alpha^2} & \frac{\partial^2}{\partial\alpha\partial\beta} \\ \frac{\partial^2}{\partial\alpha\partial\beta} & \frac{\partial^2}{\partial\beta^2} \end{pmatrix} Q_{obs} = 2 \begin{pmatrix} \overline{T_{hGHG}^2} & \overline{T_{hGHG}T_{haer}} \\ \overline{T_{hGHG}T_{haer}} & \overline{T_{haer}^2} \end{pmatrix} \tag{9}$$

The Hesse or curvature matrix $\mathbf{M}$ is characterized by its two positive eigenvalues $\lambda_1$ and $\lambda_2$ and associated eigenvectors $\mathbf{e_1}$ and $\mathbf{e_2}$, with $\lambda_1 < \lambda_2$. The extreme case of $T_{haer} \sim T_{hGHG}$, i.e. $\mathbf{M}$ is degenerate, would imply $\lambda_1 = 0$. This is not actually the case for any of the models considered here, but the two regressors are similar enough that $\lambda_1$ is close to 0. The analysis implies that the error function $E_{obs}$ forms ellipses around the minimum with two orthogonal axes that point in the directions of the eigenvectors with curvatures in these directions proportional to $\sqrt{\lambda_1}$ and $\sqrt{\lambda_2}$.

I interpret the eigenvectors $\mathbf{e_1}$ and $\mathbf{e_2}$ as the directions in $(\alpha, \beta)$ parameter space that correspond to optimal cancellation (for $\mathbf{e_1}$) and optimal reinforcement (for $\mathbf{e_2}$) of the warming effects due to GHGs and NTCFs. For the case of optimal cancellation, GHG warming and NTCF-induced cooling are at all statistically distinguishable in the observed temperature record because of a trend reversal in $SO_2$ precursor emissions in the late $20^{\text{th}}$ century (Szopa et al., 2021) causing anthropogenic aerosol-induced cooling to be on a declining trend (in absolute terms) since then, in contrast to the monotonically increasing warming since 1850 associated with GHGs. This means that variations in the contributions of both processes in this direction of optimal cancellation cause detectable variations in the temperature trend of the final 20 years of the regression fit (2001-2020 or 1995-2014, respectively) that I will relate to the trend uncertainty of the observed temperature record. This analysis will define bounds $E_{max}$ on the cost function $E_{obs}$ and consequently the regression parameters $(\alpha_1, \beta_1)$. The analysis implies that regression parameters outside these bounds yield significantly and detectably inferior regression fits.

Variations in the direction of optimal reinforcement ($\mathbf{e_2}$) by contrast produce shifts of the regression fits away from the optimum in either direction. By comparing these shifts to the uncertainty in the mean of the detrended 2001-2020 (or 1995-2014, respectively) global temperature record ($\sim$0.03 K in HadCRUT5) I find bounds on the cost function that are substantially more restrictive than the bounds associated with variations in the direction of optimal cancellation discussed above. I will therefore only present an analysis to variations in the direction of cancellation $\mathbf{e_1}$ which yields more conservative, wider error bounds.

Analogously, I define an error function $E_{hist}(\alpha, \beta)$ for the regressions to the "historical" ensemble means:

$$E_{hist}(\alpha, \beta) \equiv \sqrt{\frac{\overline{r_{hist}^2(\alpha, \beta)}}{\overline{\epsilon_2^2}}}. \tag{10}$$

where

$$r_{hist}(\alpha, \beta) = T_{hist} - \alpha T_{GHG} - \beta T_{haer} - \gamma_2 T_{hnat} - \delta_2. \tag{11}$$

The ellipses spanned by $E_{hist}$ have the same orientations and aspect ratios of the two main axes as those spanned by $E_{obs}$.

## 3 Evaluation

### 3.1 Example model calculations

As an example, figure 2 shows the result of the analysis for the HadGEM3-GC31-LL model. GHGs drive a warming of around 2 K in this model over 1850-2020 (light green line), offset by aerosol-driven cooling of around $-0.9$ K by 2020 (dark green). Natural influences explain the temporary features associated with volcanic eruptions and solar forcing (blue line). Optimal regression parameters $\alpha_2$, $\beta_2$, and $\gamma_2$ for $T_{hist}$ are close to 1 (i.e. HadGEM3-GC31-LL is nearly "additive", violet line). However, the regression against $T_{obs}$ requires substantial reductions in the parameters describing both the GHG and the aerosol influences ($\alpha_1$ and $\beta_1$), to the point that the aerosol cooling would need to be reduced by around 75%, and the GHG influence by 30%, to match the observed record (orange line).

Figure A1 contains equivalent plots for the remaining 15 models.

#### 3.1.1 When are two regression fits statistically indistinguishable?

I note that the observational record exhibits a nearly linear warming trend towards the end of the record (figure 2). Furthermore during much of the 20[th] century the aerosol-induced cooling is directly opposed to the GHG warming, but in the 1990s its trend changes sign in the HadGEM3-GC31-LL hist-aer ensemble. This trend reversal is a major reason that the GHG and aerosol influences are at all statistically distinguishable in the historical record. I thus define two regression fits to be significantly different if their 20-year trends for 2001-2020 (or 1995-2014, for CESM2, E3SM-2-0, GISS-E2-1-G, and MPI-ESM1-2-LR) differ by more than the observational uncertainties at 95% confidence in these trend ($\kappa = 6.7$ and $6.3$ mK a$^{-1}$, respectively).

I evaluate the regression fits in regression parameter space $(\alpha, \beta)$ along the lines that correspond to optimal cancellation of the warming and cooling impacts of GHGs and aerosols, respectively. This is the line spanned by the eigenvector corresponding to the smaller eigenvalue $\lambda_1$, $\mathbf{e_1}$, i.e.

$$(\alpha, \beta) = (\alpha_1, \beta_1) + c \cdot \mathbf{e_1} \tag{12}$$

This line marks the direction of maximum joint uncertainty in the regression parameters.

I plot the error functions $E_{obs}$ against the 2001-2020 (or 1995-2014) trends in the associated fits $T(\alpha, \beta) = \alpha T_{hGHG} + \beta T_{haer} + \gamma_1 T_{hnat} + \delta_1$, evaluated along the line described by eq. 12 (figure 3). By evaluating $E_{obs}$ at the two trend values that differ from the trend in the optimal solution by $\kappa$, I find model-dependent values for $E_{max}$. For all but three of the models $E_{max} \leq 1.13$. These models (BCC-CSM2-MR, FGOALS-g3, NorESM2-LM) do not simulate the trend change in the aerosol-induced cooling characterizing the other models (figure A1). The other 12 models yielding smaller, regular values for $E_{max}$ all simulate a substantial change in the rate of cooling such that their hist-aer temperature timeseries become statistically independent from their hist-GHG temperatures, and almost all exhibit warming trends during the final two decades of their hist-aer ensembles (figures 2 and A1).

## 3.2 Joint uncertainty analysis of the GHG and aerosol influences for all models

Figure 4 illustrates firstly that additivity does not extend to all models, i.e. the centres of many open ellipses are outside the "additivity rectangles". However, a subgroup of eight models does satisfy eqs. 4 and 5 (ACCESS-CM2, ACCESS-ESM1-5, CESM2, E3SM-2-0, GISS-E2-1-G, HadGEM3-GC31-LL, MIROC6, MPI-ESM1-2-LR). I note that this criterion disqualifies four models which also do not satisfy additivity in their simulated warming as expressed in figure 1(b), i.e. CanESM5, GFDL-ESM4, MRI-ESM2-0, and NorESM2-LM.

In all cases except for three models (BCC-CSM2-MR, FGOALS-g3, MPI-ESM1-2-LR) there are no substantial overlaps between the regression uncertainty ellipses for the fits to $T_{obs}$ and $T_{hist}$. This means 13 of the models have systematic differences between the simulated historical and the observed temperature evolutions large enough to show in this lack of overlap of the regression parameters. This implies irreconcilable scaling factors $\alpha$ and $\beta$ for the GHG or NTCF influences, or both.

Specifically regarding the GHG rescaling factor $\alpha$ (plotted on the horizontal axes in figure 4), for some large-ECS models including ACCESS-CM2, CanESM5, CESM2, HadGEM3-GC31-LL, and IPSL-CM6-LR, the analysis suggests that $\alpha_1 < 1$, i.e. to better match the HadCRUT5 timeseries, the GHG influences in these models need to be scaled down (Gillett et al., 2021).

Lastly, for almost all models the analysis suggests that the NTCF rescaling factor $\beta_1 < 0.6$, i.e. the filled ellipses are centred below the rectangles of additivity in figure 4. Exceptions are the MIROC6, MPI-ESM1-2-LR, and MRI-ESM2-0 models where the regression yields an NTCF influence consistent with no rescaling. Exaggeration of the NTCF influence is large and unambiguous for ACCESS-CM2, ACCESS-ESM1-5, CanESM5, E3SM-2-0, HadGEM3-GC31, and NorESM2-LM; these models all simulate at least 0.66K of cooling in their hist-aer ensembles (Gillett et al., 2021).

## 3.3 Emergent constraints for the ECS and the aerosol cooling influence

### 3.3.1 The GHG influence

Figure 5 shows the result of the regression analysis (section 2.2) for all models. The ensemble includes three models (CanESM5, CESM2, HadGEM3-GC31-LL) that have ECSs exceeding the "very likely" range given by AR6 (2 to 5 K, Forster et al., 2021, figure 5a). For these three models, the GHG correction factors $\alpha_1 < 1$ (i.e. reductions of the GHG influences would bring their historical simulations into better agreement with the observations). At the other end of the spectrum, MIROC6, with an ECS of 2.6 K, requires an increase in the GHG-induced warming by 23% to bring its historical evolution into agreement with HadCRUT5. In general, the distribution (panel a) can be approximated by ECS $\sim \alpha_1^{-1}$. Equivalently, panel (b) shows the ECS $\cdot \alpha_1$ versus $\alpha_1$. The thus "adjusted" ECSs (i.e., ECS $\cdot \alpha_1$) are now within the AR6 "likely" range (2.5 - 4 K) for all eight models that satisfy additivity, and the multi-model spread of these models' ECSs is smaller than the AR6 uncertainty range. I obtain a multi-model-mean adjusted ECS of $3.5 \pm 0.4$ K at 68% confidence (figure 5b). Replacing in the above analysis the ECSs with TCRs gives essentially the same result. Two models (CanESM5 and HadGEM3-GC31-LL) have TCRs outside the "very likely" AR6 range (1.2 to 2.4 K). Multiplying the TCR with $\alpha_1$ yields "adjusted" TCRs that are now almost all within the likely

range of AR6 (1.4 to 2.2 K) for models satisfying additivity. The multi-model mean adjusted TCR of $1.8 \pm 0.3$ K compares very well to the AR6 estimate.

### 3.3.2 The aerosol influence

The second part of this analysis concerns the aerosol-induced cooling. There is an anticorrelation (with a correlation coefficient of $-0.49$) between the cooling attributable to anthropogenic aerosol increases, as discerned from hist-aer, and the ECSs of the 16 models (figure 5e). This means that large-ECS models tend to compensate some of their GHG-induced warming by simulating a relatively large cooling due to aerosols. In other words, the biases in both properties are coupled.

Focussing here only on the eight models that satisfy additivity, their NTCF rescaling factors $\beta_1$ are in the range 0.1 to 1.25 with a mean (standard deviation) of 0.47 (0.39). In this group, the MPI-ESM1-2-LR and MIROC6 models are the only models that are consistent with no rescaling of their NTCF-induced cooling. Average adjusted aerosol-induced cooling between 1850-1899 and 2000-2014 in this group amount to $0.24 \pm 0.11$ K, when their average unadjusted cooling is $0.67 \pm 0.31$ K (both at 68% confidence). For comparison, this places my analysis at the lower end of the AR6 estimate of 0.31 (0.15 to 0.57) K (at 5 to 95% confidence) of cooling due to aerosols and ozone combined. (The AR6 range is inferred from the data accompanying figure 7.8 of Forster et al., 2021). If the models were perfect and the ozone influence was proportional to that of the aerosols, offsetting around 30% of the aerosol-induced cooling, a value $\beta_1 \approx 0.7$ would be expected, but only MIROC6 and GISS-E2-1-G get close to this. The fact that five of the other models have rescaling factors $\beta_1$ in the range 0.1 to 0.4 makes it implausible that ozone is the explanation here. Four of these models (MPI-ESM1-2-LR, MIROC6, GISS-ES-1-G, ACCESS-ESM1-5) agree within their 68% uncertainty ranges with the AR6 best estimate (0.31 K). The other four (CESM2, E3SM-2-0, ACCESS-CM2, HadGEM3-GC31-LL)require a smaller NTCF-induced cooling than that. Flynn et al. (2023) find that models that better reproduce the observed temperature evolution all simulate relatively small aerosol-induced cooling, in agreement with this study, and Gillett et al. (2021) also find small rescaling factors for the aerosol influence for several of the same models used here.

## 4 Discussion and conclusions

Mismatches between the observed global-mean surface temperature and CMIP6 "historical" simulated temperature have been documented before and attributed, in some cases, to a deficient simulation of aerosol-related cooling (Andrews et al., 2020; Flynn and Mauritsen, 2020; Smith and Forster, 2021; Golaz et al., 2022; Flynn et al., 2023). Here I exploit these mismatches to derive scaling factors for the GHG-induced warming and the aerosol-related cooling on temperature that in a hypothetical model would bring the simulated historical temperature into optimal agreement with the HadCRUT5 climatology. I then relate these scaling factors to the warming attributable to GHGs, on the one hand, and on the other hand to the cooling attributable to anthropogenic NTCFs. The GHG scaling factors very approximately follow an inverse relationship with the ECSs of the models, such that the products of the ECSs and the scaling factors are in better agreement with the AR6 evaluation of the planetary ECS than the modelled ECSs themselves. Particularly for three large-ECS models with ECSs outside the AR6 "very

likely" range (Forster et al., 2021), this adjustment brings these ECSs into agreement with the AR6 estimate. Essentially the same holds for the TCR.

These results are consistent with quantifications of the aerosol and GHG influences based on energy balance calculations (e.g. Storelvmo et al., 2016; Smith and Forster, 2021; Smith et al., 2021) but arrived at using an independent approach. Smith and Forster (2021) use energy budget constraints to find that despite reductions in historical aerosol and GHG forcing from CMIP5 to CMIP6, stronger climate feedbacks in CMIP6 models, which are reflected in the increased ECSs in CMIP6 models, cause both stronger aerosol cooling and from 1990 increased GHG warming. This analysis generally confirms their results and but shows that a tendency to overestimate aerosol-induced cooling pertains not just to the high-ECS models but to many of the moderate-ECS models as well. Only for one model (MPI-ESM1-2-LR) I find a rescaling factor larger than 1 (Gillett et al., 2021). At 0.26 K the mean aerosol cooling simulated by MPI-ESM1-2-LR is the second-smallest in the ensemble. Storelvmo et al. (2016) employ a purely observations-based approach to find that aerosols masked approximately 1/3 of the GHG-induced warming since the 1960s, leaving a TCR of 2±0.8K. Again, my results are consistent with but largely independent of their results. Smith et al. (2021) use an energy balance approach constrained by CMIP6 input data to quantify the ECS and the TCR at 3.1 and 1.8 K, respectively, in agreement with my results, and give a very wide uncertainty range of $-1.8$ to $-0.5$ Wm$^{-2}$ since 1750. Applying a conversion factor of 0.5 K W$^{-1}$ m$^2$ (Forster et al., 2021) yields 0.25 to 0.9 K of cooling due to aerosol. Furthermore adjusting for some anthropogenic aerosol increase between 1750 and 1850-1899, and accounting for some offset by ozone, my result is consistent but near the low end of this range.

Hodnebrog et al. (2024), in a recent study, attribute much of the observed recent increase in the Earth's energy imbalance to a reduction in the anthropogenic aerosol forcing. While the methodology used by these authors differs from my approach, this finding is likely inconsistent with the conclusion arrived at here that aerosols, in the adjusted multi-model mean, exert a substantially smaller influence on climate than the ever-increasing GHGs. It also appears inconsistent with the plain results of the hist-aer and hist-GHG simulations (figures 2 and A1), where the warming attributable to aerosols over 2001-2019 is generally smaller than that simulated in the hist-GHG ensembles over the same period. This is even before this warming is scaled down, as is laid out above. More research is required to reconcile these findings.

I find an anticorrelation between the total simulated aerosol-induced cooling since 1850 and the ECSs of the models, suggesting that both quantities are coupled and that there could be a compensation of errors between these two processes. The eight models considered here, that satisfy additivity, span a considerable range in simulated aerosol-induced cooling for 1850-1899 to 2000-2014, namely 0.3 to 1.2 K, with a mean (standard deviation) of 0.67 K (0.31 K). Applying the correction factor $\beta_1$ reduces the range to 0.04 to 0.36 K with a mean (standard deviation) of 0.24 (0.11) K. This is now interpreted to mean the cooling due aerosols between 1850-1899 and 2000-2014, offset by warming due to ozone in the same period.

There are several limitations to the analysis presented here: The first is that the hist-GHG experiment quantifies the responses of the climate models to all GHGs in combination, whereas the ECSs and TCRs are expressions of the sensitivity of climate only to $CO_2$ increases. I have shown that there are near-perfect proportionalities and high degrees of correlation (0.80 and 0.87, respectively) between the warmings simulated in hist-GHG and the ECSs and TCRs in the 16 models used here (figure

1), suggesting that the substantial model diversities that exist for these quantities are due to the same processes, i.e. climate feedbacks e.g. due to cloud adjustments that are not sensitive to the detailed properties of the driving GHGs.

A further, more fundamental limitation is that the models do not respond perfectly additively to GHG and aerosol forcing. This is expressed in deviations from 1 for the $\alpha_2$, $\beta_2$, and $\gamma_2$ parameters in equation 3 (figure 4). The non-additivity is due to a variety of reasons, including forcings not included in the analysis (such as land use and ozone changes). I have tested the sensitivity of the results to including in the analysis the impact of warming due to ozone changes, by using the "hist-totalO3" simulations that were produced under DAMIP, and by correspondingly expanding equations 2 and 3 by a term for the temperature anomalies simulated under this experiment. Five models (CanESM5, GISS-E2-1-G, HadGEM3-GC31-LL, MIROC6, and MPI-ESM1-2-LR) have completed this experiment but two of these (CanESM5, MPI-ESM1-2-LR) are "non-additive" in the expanded regression model. Ozone changes are the most important anthropogenic radiative forcing agent after those considered here (GHGs and aerosols, Forster et al., 2021). Repeating the analysis based on just GISS-E2-1-G, HadGEM3-GC31-LL, and MIROC6, but with a term added to the regression models for ozone-induced warming, I obtain quite similar values for the ECS ($3.2\pm1$ K), TCR ($1.8 \pm 0.4$ K), and aerosol-induced cooling ($-0.29\pm0.11$ K. This suggests that ozone forcing is neither the leading explanation for the non-additivity nor for the small aerosol-induced cooling found here. More models completing the hist-totalO3 simulations would help.

A further reason for non-additivity could be a substantial influence of random variations on the regression. I note that the model with the joint smallest hist-aer ensemble size (GFDL-ESM4) exhibits too substantial a non-additivity to be included in the multi-model averages (section 3.3), and the models with the best additivity (HadGEM3-GC31-LL, MIROC6, MPI-ESM1-2-LR) have all contributed large ensembles. However there is not a consistent association of additivity with ensemble sizes. For example, CanESM5 has large ensembles for all experiments but still exhibits substantial non-additivity. Quite a few models have ensemble sizes of 3 for their hist-aer or hist-GHG ensembles, but this group includes some models with good and poor additivity. In some cases, the non-additivity can be traced to an extremely small eigenvalue $\lambda_1$ of the covariance matrix $\mathbf{M}$: For HadGEM3-GC31-LL, as an example of an "additive" model, I find $\lambda_1 = 0.02$. Examples of poor additivity include FGOALS-g3 ($\lambda_1 = 8 \cdot 10^{-4}$), IPSL-CM6A-LR ($\lambda_1 = 3 \cdot 10^{-3}$), and NorESM2-LM ($\lambda_1 = 6 \cdot 10^{-3}$). However, the group of additive models also includes MIROC6 ($2 \cdot 10^{-3}$). So near-degeneracy of $\mathbf{M}$ also does not completely explain why some models exhibit non-additivity. Other factors may come into play, including that some models simply respond non-linearly to the applied forcings, or even that errors exist in the experimental setups. It is beyond the scope of this paper to fully diagnose these occurrences of non-additivity. However, removing such models from the emergent-constraint calculations of section 3.3 substantially improves model consensus.

Several models indicating that large reductions in aerosol cooling would be beneficial for bringing the simulated historical temperature record into better agreement with observations, including ACCESS-ESM1-5, E3SM-2-0, HadGEM3-GC31-LL, and MRI-ESM2-0, all have $\beta_2 > 0.6$, i.e. these models behave relatively additively, and the inference that exaggerated historical aerosol-induced cooling contributes substantially to errors in the simulations of global-mean temperature by these models is quite well founded.

Bellouin et al. (2020) review the constraints posed by observed temperature on the effective radiative forcing of aerosols. Applying a conversion factor of $0.5 \text{ K W}^{-1} \text{ m}^2$ (Forster et al., 2021) I translate their effective radiative forcing estimate into a cooling influence over the historical period. In their discussion, which considers both direct and indirect aerosol effects on radiation, the best-estimate uncertainty interval of aerosol radiative forcing ($-1.6$ to $-0.35 \text{ Wm}^{-2}$, translated into approximately 0.18 to 0.8 K of aerosol-induced cooling) includes most models considered here for both their unadjusted and adjusted aerosol-induced cooling, although the best-estimate multi-model mean cooling due to NTCFs inferred here of 0.24 K is near the low end of this range. Accounting for the influence of ozone (Forster et al., 2021), this translates into a best estimate of 0.38 K, which better compares to Bellouin et al. (2020)'s headline estimate. As noted above, my estimate for the cooling due to NTCFs is skewing slightly smaller than the one in AR6 but with overlapping uncertainty ranges.

The results qualitatively confirm Smith and Forster (2021) who find that excessive cooling due to aerosols in 1960-1990 causes cold biases in this period in many CMIP6 "historical" simulations. My analysis leaves open the question whether the excessive response to aerosol forcing in most models is due to too much aerosol being produced in these models (i.e. a problem with the CMIP6 forcing data), or whether the internal model physics of aerosol-radiation and aerosol-cloud interactions is flawed. The fact that this behaviour is common to most models suggests the former is a likely factor.

Figure 3 shows that the regression coefficients versus the HadCRUT5 temperature, $\alpha_2$ and $\beta_2$, are subject to substantially larger uncertainties than those versus the ensemble-mean simulated temperatures, $\alpha_1$ ad $\beta_1$. This is a reflection of the larger noise associated with observed temperature compared to an ensemble mean surface temperature. Despite this larger uncertainty, the analysis does indicate, for most models, differences between these two parameter pairs that are irreconcilable within the statistical uncertainties. In a follow-on paper I will investigate this aspect further using a probabilistic approach.

In summary, I have used a reconstruction of global-mean merged surface / surface-air temperature and DAMIP and historical simulations by 16 contemporary climate models to derive constraints for the GHG-induced warming and the aerosol-induced cooling, by far the leading influences driving global warming. Using an emergent constraint approach, I derive a "corrected" ensemble-mean equilibrium climate sensitivity of about $3.5 \pm 0.4$ K and a corrected TCR of $1.8 \pm 0.3$ K (both at 68% confidence), in excellent agreement with the AR6 estimates but with reduced uncertainties (Forster et al., 2021). For eight models with relatively good additivity, I find that reductions in the NTCF-induced cooling, along with some reductions in the GHG-induced warming for models with large ECSs, would bring their historical simulations into better agreement with the observational record. The results presented here highlight ongoing difficulties with correctly simulating climate feedbacks in global models. Substantial, systematic, and nearly community-wide issues with representing historical global surface temperature reduce confidence in quantitative projections of global warming by models affected by these problems. Interestingly, at least some CMIP3 models were consistent with observations without any need for rescaling the aerosol and GHG signatures (Stone et al., 2007b), including for the precursor of CESM2 (Stone et al., 2007a). This may suggest that at least for this model development occurring in the intervening time has introduced this problem.

The analysis is limited by the substantial anticorrelation between the GHG and the aerosol global-mean warming signatures. I anticipate that as anthropogenic aerosol production continues, as projected, to decline in the future (Lee et al., 2021), the anti-

correlation between GHG-induced warming and aerosol-induced cooling will reduce, allowing for a more confident attribution

375  of their respective roles in driving global warming.

*Data availability.* All model data used here have been downloaded from the Earth System Grid Federation, e.g. https://esgf-node.llnl. gov/search/cmip6/. HadCRUT5.0.1.0 data were obtained from http://www.metoffice.gov.uk/hadobs/hadcrut5 on 1 July 2023 and are © British Crown Copyright, Met Office [2019], provided under an Open Government License, http://www.nationalarchives.gov.uk/doc/open-government-licence/version/3/.

380  *Code and data availability.* Code and data to generate the figures of this paper are at https://doi.org/10.5281/zenodo.11366923.

## Appendix A: Regression fits for the remaining models

For models not represented in figure 2, the regression fits are given in figure A1.

### A1

*Author contributions.* OM conceived of and conducted the analysis and wrote the paper.

385  *Competing interests.* The author declares no competing interests.

*Acknowledgements.* I acknowledge fruitful discussions with Dáithí Stone. I acknowledge the World Climate Research Programme, which, through its Working Group on Coupled Modelling, coordinated and promoted CMIP6. I thank the climate modeling groups for producing and making available their model output, the Earth System Grid Federation (ESGF) for archiving the data and providing access, and the multiple funding agencies who support CMIP6 and ESGF. I acknowledge the UK MetOffice for providing the HadCRUT5 data. I acknowledge two
390  reviewers for their thoughtful, constructive comments that have helped improve the paper.

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

**Table 1.** DAMIP models, ensemble sizes of the experiments, literature ECS values of the models, and references for the ECS values and the model data. MPI-ESM1-2-LR actually has not completed hist-nat. For this model the hist-nat temperature change is inferred from its "hist-sol" and "hist-volc" ensembles (Gillett et al., 2016).

| Model | Sizes of ensembles (historical/hist-aer/hist-GHG/hist-nat/ssp245) | ECS (K) | TCR (K) | Reference for ECS and TCR | References for data |
|---|---|---|---|---|---|
| ACCESS-CM2 | 5/3/3/3/10 | 4.7 | 2.1 | Meehl et al. (2020) | Dix et al. (2019a, b, 2020a, b, c) |
| ACCESS-ESM1-5 | 40/3/3/3/40 | 3.9 | 2.0 | Meehl et al. (2020) | Ziehn et al. (2019a, b, 2020a, b, c) |
| BCC-CSM2-MR | 3/3/3/3/1 | 3.0 | 1.7 | Meehl et al. (2020) | Wu et al. (2018, 2019a, b, c); Xin et al. (2019) |
| CanESM5 | 25/15/25/25/25 | 5.6 | 2.7 | Meehl et al. (2020) | Swart et al. (2019a, b, c, d, e) |
| CESM2 | 11/2/1/2/0 | 5.2 | 2.0 | Meehl et al. (2020) | Danabasoglu (2019a, b, c, 2020) |
| CNRM-CM6-1 | 21/10/10/10/6 | 4.8 | 2.1 | Meehl et al. (2020) | Voldoire (2018, 2019a, b, c, d) |
| E3SM-2-0 | 11/5/5/5/0 | 4.0 | 2.4 | E3SM (2022a) | E3SM (2022b, c, d, e) |
| FGOALS-g3 | 5/1/3/3/4 | 2.9 | 1.5 | Scafetta (2023) | Li (2019, 2020a, b, c, d) |
| GFDL-ESM4 | 3/1/1/3/3 | 2.6 | 1.6 | Meehl et al. (2020) | Krasting et al. (2018); Horowitz et al. (2018a, b, c); John et al. (2018) |
| GISS-E2-1-G | 11/5/18/5/0 | 2.7 | 1.8 | Meehl et al. (2020) | NASA/GISS (2018a, b, c, d) |
| HadGEM3-GC31-LL | 55/55/55/57/55 | 5.6 | 2.6 | Meehl et al. (2020) | Ridley et al. (2019); Jones (2019a, b, c); Good (2019) |
| IPSL-CM6A-LR | 33/10/10/10/11 | 4.6 | 2.3 | Meehl et al. (2020) | Boucher et al. (2018a, b, c, d, 2019) |
| MIROC6 | 50/10/50/50/50 | 2.6 | 1.6 | Meehl et al. (2020) | Tatebe and Watanabe (2018); Shiogama (2019a, b, c); Shiogama et al. (2019) |
| MPI-ESM1-2-LR | 38/30/30/30/0 | 3.0 | 1.8 | Meehl et al. (2020) | Wieners et al. (2019); Müller et al. (2019a, b, c, d) |
| MRI-ESM2-0 | 10/3/5/4/5 | 3.2 | 1.6 | Meehl et al. (2020) | Yukimoto et al. (2019a, b, c, d, e) |
| NorESM2-LM | 3/1/3/3/3 | 2.5 | 1.5 | Meehl et al. (2020) | Seland et al. (2019a, b, c, d, e) |

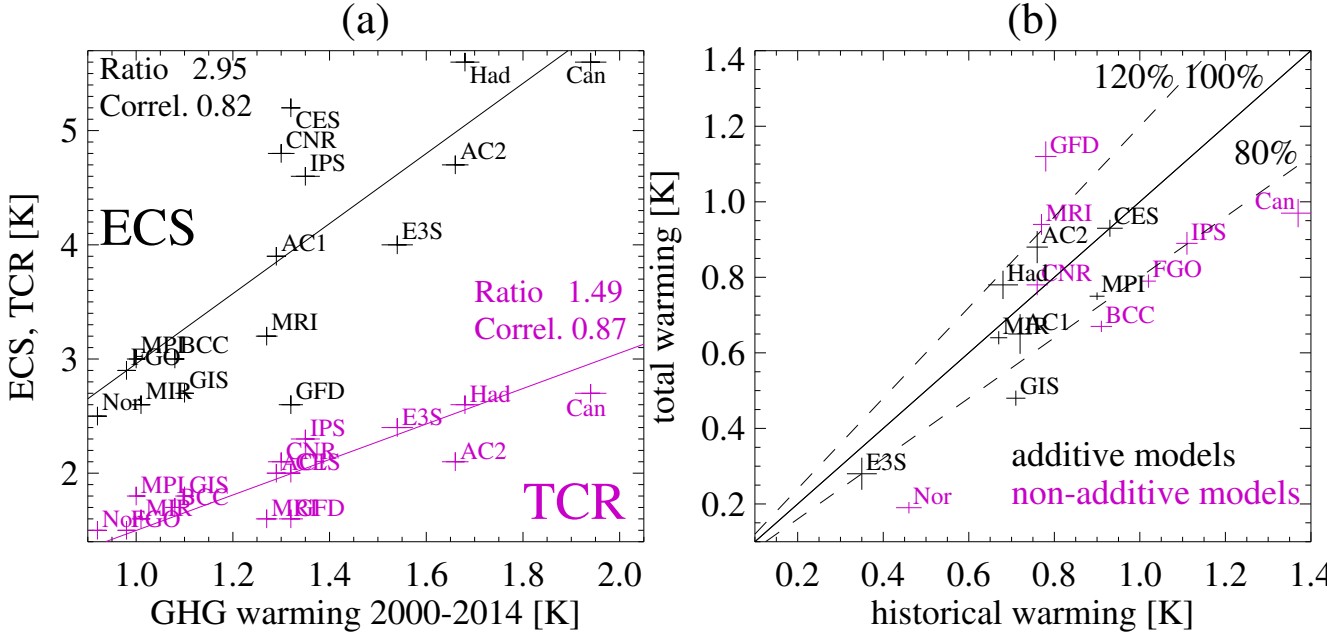

**Figure 1.** (a) Simulated global- and annual-mean warming for 2000-2014 in the hist-GHG ensembles of the DAMIP models, relative to the 1850-1899 mean, versus their ECSs (black) and TCRs (violet, all in K). The width of the horizontal lines corresponds to $\mathrm{std}\,(T_i)/\sqrt{15}$, where the $T_i$ are the annual-mean temperatures for 2000-2014. Solid lines: best-estimate proportional fits. The models' names are abbreviated to three characters. AC1 = ACCESS-ESM1-5. AC2 = ACCESS-CM2. Also stated are the best-fit proportionality constants and correlation coefficients. (b) Simulated global-mean warming between 1850-1899 and 2000-2014 in the "historical" ensembles versus the sum of these warmings simulated in the respective hist-GHG, hist-aer, and hist-nat ensembles. The solid line marks the diagonal, dashed lines the 80 and 120% lines. The lengths of the bars in both directions correspond to the statistical uncertainties at 68% confidence. Models marked in violet are excluded from the multi-model mean calculations because they do not satisfy the additivity constraints (eqs. 4, 5).

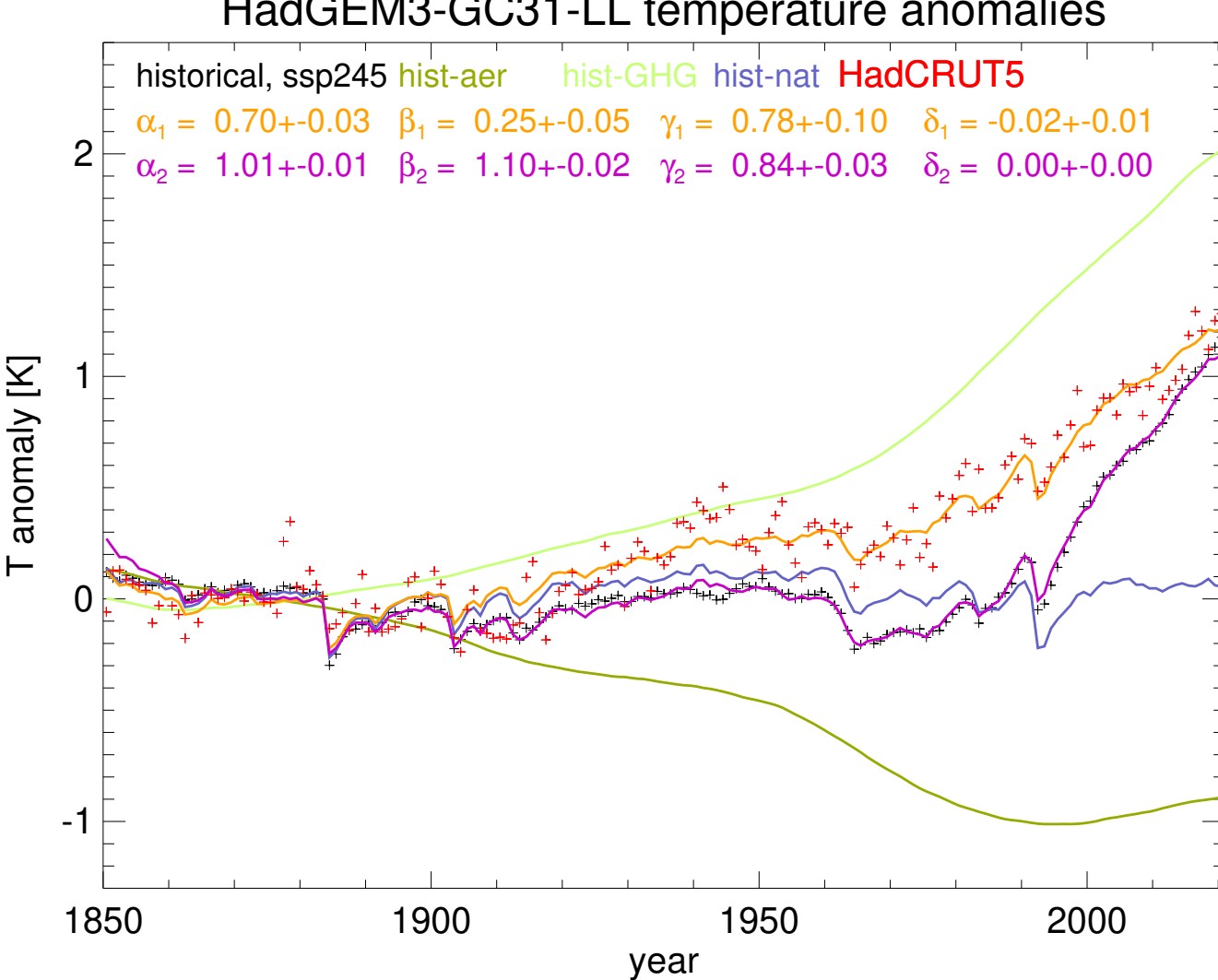

**Figure 2.** Ensemble-, global-, and annual-mean temperature anomalies relative to the 1850-1899 average for HadGEM3-GC31-LL. Black symbols and dark green, light green, and blue lines: The DAMIP and "historical"/SSP245 ensemble means as indicated. The hist-GHG and hist-aer temperature evolutions have been smoothed using a 15-year boxcar filter. Violet: Optimal regression fits to $T_{hist}$ following equation 3. Orange: Optimal regression fits to $T_{obs}$, the HadCRUT5 reconstruction, following equation 2. Red symbols: HadCRUT5 (Morice et al., 2021). The regression coefficients $\alpha_1$, $\beta_1$, $\gamma_1$, $\delta_1$, $\alpha_2$, $\beta_2$, $\gamma_2$, and $\delta_2$ that are stated in orange and violet are as defined in equations 2 and 3.

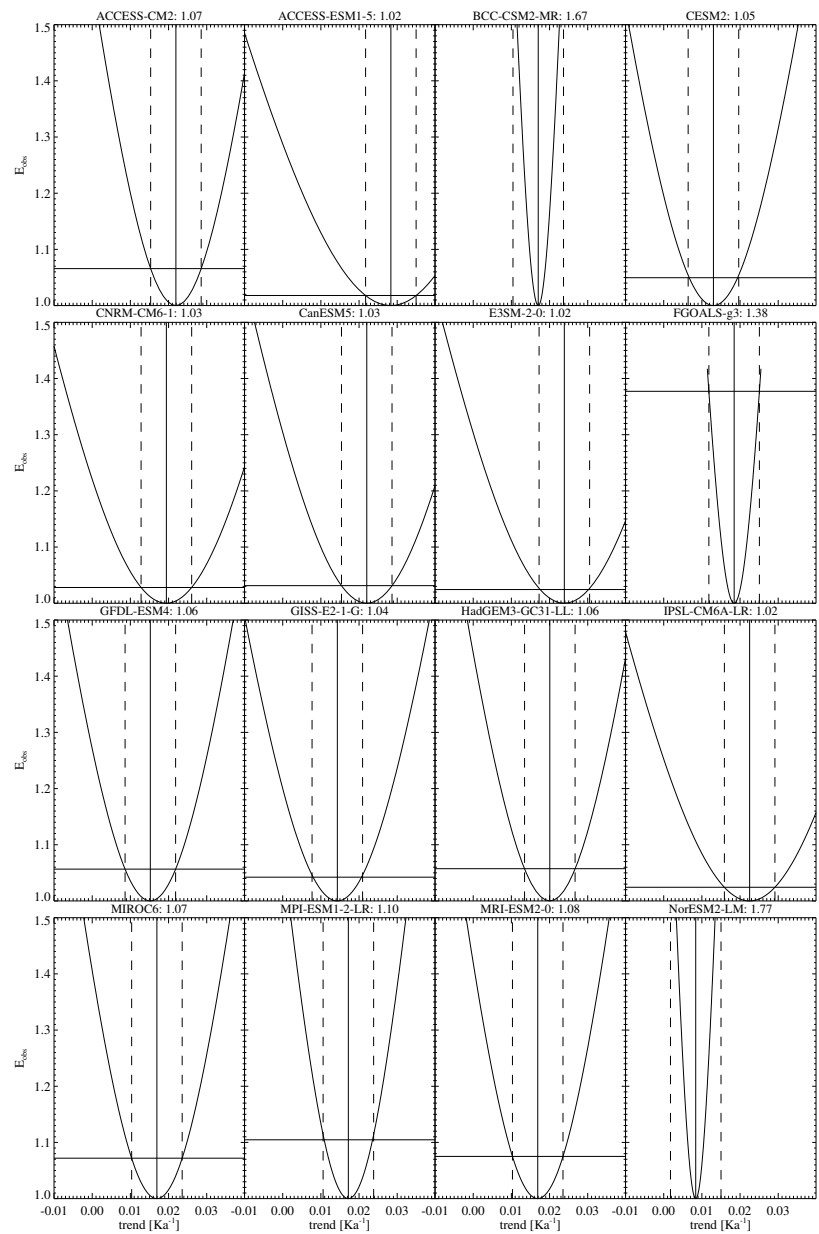

**Figure 3.** Error function $E_{obs}$ as a function of the 2001-2020 (for CESM2, E3SM-2-0, and GISS-E2-1-G: 1995-2014) temperature trends in the regression fits to $T_{obs}$ (eq. 2), for the DAMIP models. The trends are evaluated along the line in $(\alpha, \beta)$ parameter space described by equation 12. Vertical solid line: Trend in the optimal fit (that minimizes $E_{obs}$). Vertical dashed lines: Trends in the sub-optimal fits that differ from the optimal trend by the observational trend uncertainty $\kappa$. Horizontal line: Value of the error function corresponding to this trend uncertainty. The number in the titles is the value of the error function at these points.

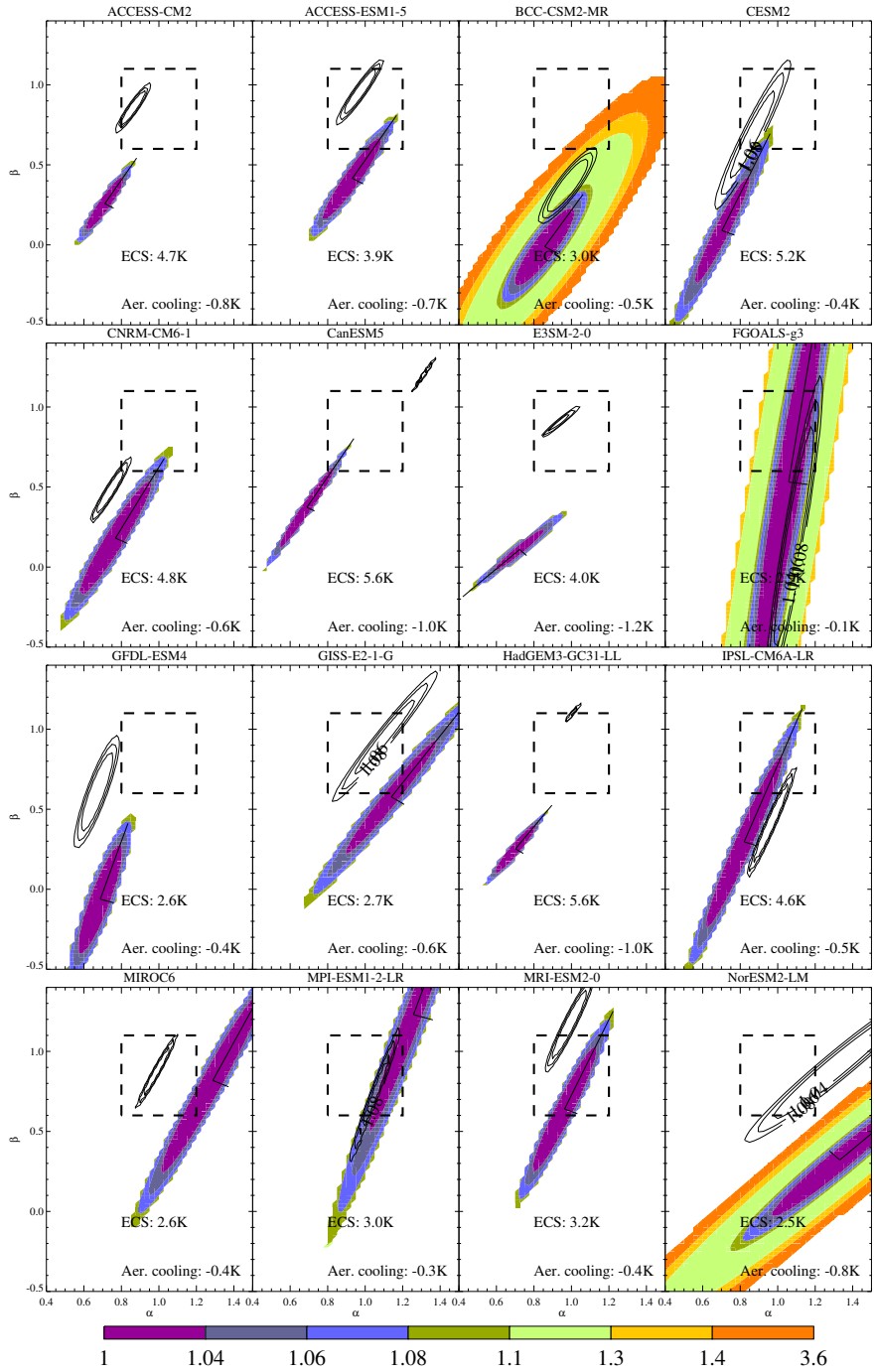

**Figure 4.** Error functions $E_{obs}$ (colours, eq. .7) and $E_{hist}$ (contours, eq. 10) where these functions are smaller than $\max(E_{max}, 1.1)$, for the 16 DAMIP models. Rectangle: Window of additivity defined by eqs. 4 and 5. "Aer. cooling" is the global-, ensemble-, and annual-mean cooling for 2000-2014 relative to 1850-1899 as simulated in the models' hist-aer ensembles. "ECS" is as in table 1.

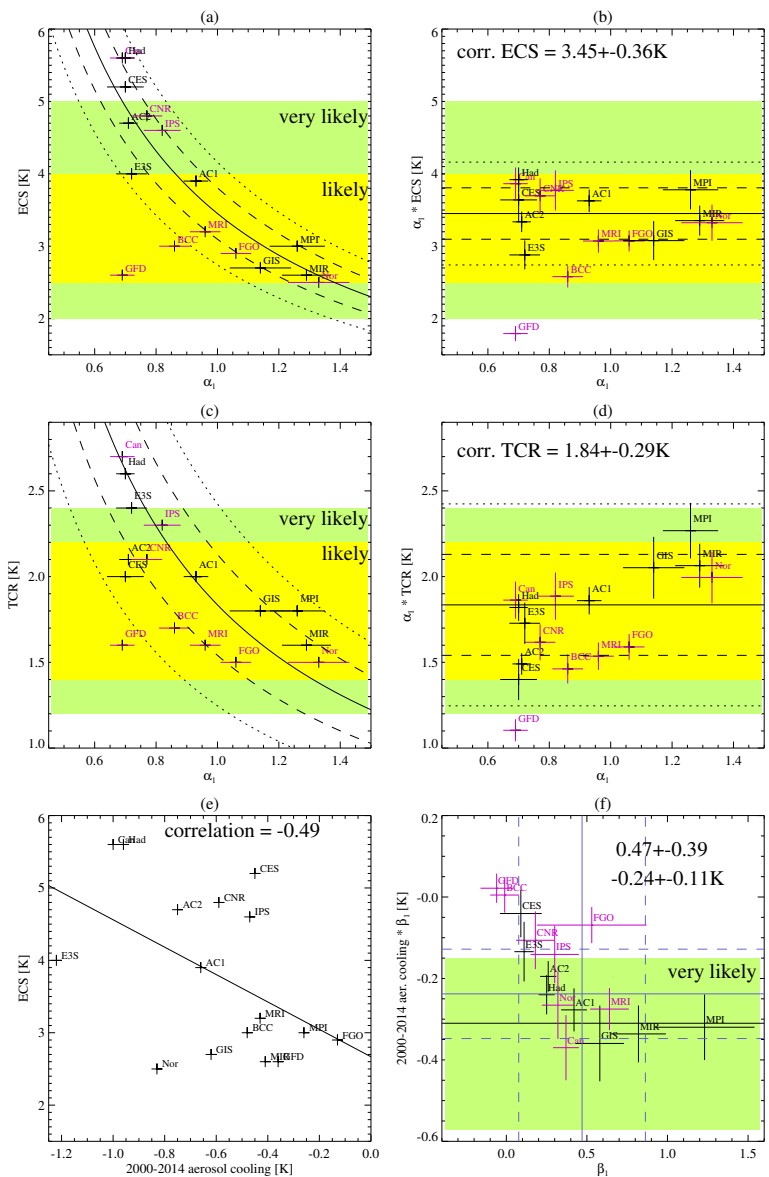

**Figure 5.** (a) The models' ECSs (K) versus $\alpha_1$ (equation 2). The lengths of the horizontal lines depict the regression uncertainties at 68% confidence. Solid line: Regression fit assuming ECS·$\alpha_1$ =const. Dashed and dotted lines: Uncertainty ranges at 68 and 95% confidence. The yellow and green regions are the likely (i.e. 66% confidence) and very likely (90%) ECS intervals assessed by AR6 (Forster et al., 2021). (b) Same as (a) but for ECS·$\alpha_1$ versus $\alpha_1$. The solid, dashed and dotted lines are the mean and the 68 and 95% confidence intervals. (c) and (d): Same as (a) and (b) but for the TCR. (e) The tabulated ECSs versus cooling simulated in the hist-aer ensembles for 2000-2014 relative to 1850-1899. Solid line: Best linear fit. (f) The aerosol-induced cooling taken from hist-aer times $\beta_1$ (K) versus the correction factors to aerosol cooling $\beta_1$ (equation 2). Blue solid and dashed lines denote the means and 68% confidence limits of both quantities. The black line and the green box are the AR6 best estimate and the 5 to 95% uncertainty range for the temperature change due to NTCFs (figure 7.8 of Forster et al., 2021). The numbers in black denote the means and standard deviations of $\beta_1$ and of aerosol-induced cooling times $\beta_1$, in K. The models marked in violet are excluded from this averaging.

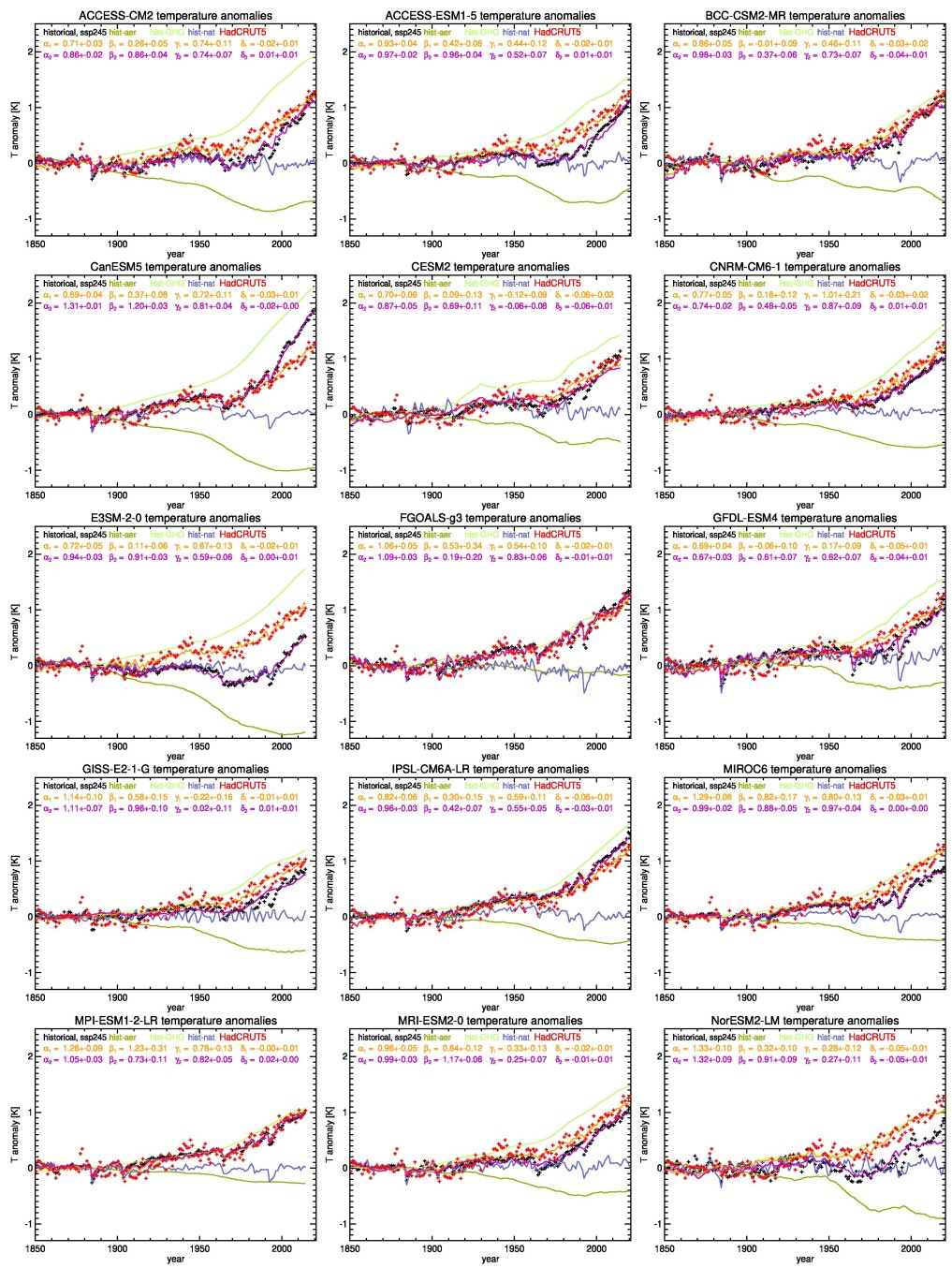

**Figure A1.** Same as figure 2 but for the ACCESS-CM2, ACCESS-ESM1-5, BCC-CSM2-MR, CanESM5, CESM2, CNRM-CM6-1, E3SM-2-0, FGOALS-g3, GFDL-ESM4, GISS-E2-1-G, IPSL-CM6A-LR, MIROC6, MPI-ESM1-2-LR, MRI-ESM2-0, and NorESM2-LM models.