# Peer review of "Using historical temperature to constrain the climate sensitivity, the transient climate response, and aerosol-induced cooling"

_EGUsphere, 2023_

## Author Comment (AC1)

Reviewer #1:

I thank the reviewer for thoughtful and constructive comments.

The author aims to constrain both the climate sensitivity and the aerosol-induced cooling (between pre-industrial to present-day) from the time evolution of the global mean surface temperature. The constraint comes from the decoupling between the greenhouse gas and aerosol forcings that occurred between 1980 and 2000 and its impact on the historical temperature record. The author finds a smaller aerosol cooling effect than predicted in the models but also a smaller aerosol cooling effect than generally found in other detection and attribution studies. While this manuscript is complementary to previous studies and it represents an interesting contribution to the literature, I doubt it is the end of the story.

I doubt that too.

Furthermore I have a number of major comments that would require clarifications and most likely further analysis. Overall I recommend major revisions to the manuscript before it can be considered for publication in Atmospheric Chemistry and Physics.

Major comments

1/ The author should do a much better job at citing and describing previous work on the topic (e.g. doi: 10.1038/s41558-020-00965-9, 10.1175/JCLI-D-19-0091.1, 10.1038/ngeo2670 but I am sure there are other references as well). There is a large variety in approaches. Storelvmo et al. used a multivariate fingerprint of the aerosol impact on the climate system. Charles et al. relied on time variations of the ocean heat content. Gillet et al. used not only the magnitude but also the pattern of the aerosol cooling to constrain the climate sensitivity and forcing. The uncertainties remain large but available studies generally find a larger contribution of the aerosols (relative to greenhouse gases) than found in this manuscript. For instance Gillett et al. (2021) concluded that "Greenhouse gases and aerosols contributed changes of 1.2 to 1.9 °C and −0.7 to −0.1 °C, respectively, and natural forcings contributed negligibly." The author should discuss or at least speculate why his conclusions are somewhat different from those of past studies.

> **Commented [OM1]:** Gillett et al., NCC, 2021
>
> **Commented [OM2]:** Elodie, C., Meyssignac, B., & Ribes, A. (2020). Observational constraint on greenhouse gas and aerosol contributions to global ocean heat content changes. *Journal of Climate, 33*(24), 10579-10591. doi:https://doi.org/10.1175/JCLI-D-19-0091.1
>
> **Commented [OM3]:** Storelvmo, T., Leirvik, T., Lohmann, U. *et al.* Disentangling greenhouse warming and aerosol cooling to reveal Earth's climate sensitivity. *Nature Geosci* **9**, 286–289 (2016). https://doi.org/10.1038/ngeo2670

I have now strengthened the discussion of the literature. Previous authors including Gillett et al (2021) have found a highly uncertain aerosol-induced cooling that my results are generally consistent with. The main contribution of this paper is that I explicitly consider additivity and disqualify models that do not satisfy additivity. This results in less model disagreement. Previous studies (e.g. Gillett et al. 2021) have generally assumed additivity, have included all available models, and as a result have derived quite uncertain results. This is now discussed in some detail.

2/ The author compares the global surface air temperature (variable tas) from the models to the global-mean surface temperature from HadCRUT5 but fails to recognize that these are two different things. HadCRUT5 is actually a mix of SST over the ocean and surface air temperature over land. It should be noted that the trends in GMST and GSAT are somewhat different because of the contributions from regions of melting sea ice. Indeed GSAT increases a lot in places where sea ice melts while SST is hardly affected. Furthermore the kink in GMST and GSAT due to aerosols may not be synchronous. Both IPCC and Gillett et al (2021) moved away from GMST to embrace GSAT. I

strongly recommend that the author repeats his analysis with GSAT observations (such as provided by the Berkeley Earth project) to avoid the inconsistency in the way temperature trends are estimated between models and observations.

To improve consistency with HadCRUT5, I now use as the basis for comparison with HadCRUT5 an "amalgamated" temperature product which is SST over the ocean and SAT over land. (This is how HadCRUT5 is constructed.) The results only differ marginally from what I had originally. I understand that this is done because surface-air temperature is what land-based met stations routinely measure, whereas historic ship-based observations were usually of SST. Given the insensitivity of the results with regard to the choice of ts or tas, only HadCRUT5 is used here, and model data are now used in a way that is consistent with this choice.

3/ It is annoying and worrying that the total warming (from the sum of the three experiments) differs so much from the historical warming given that climate models are generally thought to be largely additive for small perturbations. Is the plot in Fig 1b for the ensemble mean?

Indeed these numbers reflect the ensemble means. Actually several small changes to the regression model (namely the introduction of an "intercept" $\delta$ in the new Eq. 2, which accounts for small problems with the normalization of the temperature datasets and makes the regression symmetric, also smoothing of the hist-ghg and hist-aer ensemble means) for most models improves this situation. Also I now re-interpret the terms $\sim T_{haer}$ to stand for the total influence of near-term climate forcers (NTCFs, i.e. aerosols and ozone). This means the expected value for $\beta2$ is now < 1. I now discuss that eight models satisfy additivity (i.e. (a) $|\alpha_2 - 1| <= 0.2$ and (b) $0.6 <= \beta_2 <= 1.1$). In the calculation of the corrected ECS, TCR, and NTCF-induced cooling only those models are used that satisfy both (a) and (b) are considered. Selection of only those models yields substantially smaller statistical uncertainties for the resultant best-estimate correction factors because some models yielding relatively extreme correction factors are then systematically excluded. I think exclusion of the non-additive models is the main reason for the reduced model uncertainty found here, relative to other papers. However, the flip side of the coin is that now this is based on a relatively small number of models (i.e. 8). I think this is the smaller price to pay, relative to including models that are unsuitable for this attribution study because they are not additive.

How large is the ensemble?

Ensemble sizes are given in table 1.

How significant is the non-additivity?

For some models the non-additivity is substantial. Those models are now not considered in calculations of best-estimate correction factors, TCRs, ECSs, and adjusted NTCF-induced cooling. More models satisfy additivity for the GHG influence than the aerosol influence. I have therefore introduced a more stringent condition ($|\alpha_2 - 1| < 0.2$) for the GHG additivity (which removed 5 models) than for the aerosol influence ($0.6 <= \beta2 <= 1.1$). This removes 8 models. 8 models remain that satisfy both criteria. Consideration of models that satisfy additivity considerably reduces uncertainties for all multi-model means calculated here. Figure 1b indicates that the 8 models retained for the analysis all exhibit a relatively good correspondence between simulated "historical" warming and warming summed up over the three sensitivity experiments. The eight excluded models comprise four (GFDL-ESM4, MRI-ESM2, CanESM5, NorESM2-LR) where this total warming is outside the 80-120% envelope of simulated historical warming.

Can it be explained by natural variability or missing terms in the total warming?

The main missing term is the ozone forcing. This is now considered in two ways: Firstly, I now provide an emergent-constraint analysis of warming due to the combination of aerosols and ozone, however based only on the hist-aer experiment. The underlying assumption here is that the warming due to ozone is proportional to that due to aerosols. According to IPCC (figure 7.8) this is a pretty good assumption. Throughout 1850-2019 ozone has consistently offset ~1/3 of the cooling due to aerosols, a little more in the beginning (when both forcings were small) and in the end when aerosols are decreasing). This motivates that I now have an expected value of $\beta_2$ of ~0.7.

In addition, I have repeated the analysis for the 5 models that have conducted the hist-totalO3 experiment. For CanESM5, the analysis confirms that this model is substantially non-additive. For MIROC6, $\beta_2$ goes from 0.76 to 1.01 (in agreement with the above expectation), but in this model, the ozone forcing hardly produces any sustained warming, so this may be fortuitous. In HadGEM3 and GISS-E2-1-G, the coefficients remain largely unchanged. MPI-ESM1-2-LR is non-additive when ozone forcing is included, because warming due to ozone is very nearly of the same size and directly opposed to aerosol-induced cooling in this model, making them impossible to separate.

So with the presently available hist-totalO3 simulations, a robust analysis of the influence of ozone on global-mean temperature remains very difficult.

And if so what is the impact on the author's analysis? If the departure of $\alpha_2$ and $\beta_2$ from unity is due to natural climate variability (which I suspect), then I do not see the rationale for normalizing $\alpha_1$ and $\beta_1$ with $\alpha_2$ and $\beta_2$ as done in Eq. 3. If the departure of $\alpha_2$ and $\beta_2$ from unity is instead due to missing forcing terms, then should the missing term not be inserted in Eq 1? Fig 4 shows that the departure of the black ellipses from the (1,1) point is quite generalized. If it was due to climatological noise, then should we not expect values that are both smaller and larger than 1? Here we have mostly values smaller than 1 (especially for $\beta_2$). Such low values question the validity of the framework. I note that the author discusses this issue in the conclusion section (line 227ff), however this comes far too late in the manuscript. The fact that $\beta_2 < 0.5$ for several model does not only "question the suitability of these models for attribution", it questions the suitability of the method. I would strongly recommend the author to investigate further the reason for non-unity values of $\alpha_2$ and $\beta_2$ and its impact on the analysis. This is a show-stopper in my opinion.

As noted, there is no conclusive evidence that the non-additivity is due to missing forcing terms. I cannot straightforwardly expand the regression model because other experiments in DAMIP are tier-2, and few models have conducted these simulations.

I agree with the reviewer that the best course of action w.r.t. non-additivity is not to include such models in calculations of multi-model averages. This is what I have done in the revised version of the manuscript.

The main impact is the introduction of a straightforward criterion to distinguish between suitable and unsuitable models. Based on this criterion uncertainty ranges for both the GHG-induced warming (ECS, TCR) as well as the aerosol-induced cooling are reduced versus other assessments. Also the best-estimate aerosol-induced cooling is smaller than found in other studies, though well within their uncertainty ranges.

4/ How do uncertainties in $T_{obs}$, not accounted for in the current analysis, affect the results?

We have replaced HadCRUT5 with Berkeley Earth, an alternative global temperature dataset (the variant where SSTs over sea ice are inferred from open water around sea ice). Berkeley Earth uses the same HadSST4 dataset for sea-surface temperature as HadCRUT5. Hence there are no qualitative

differences in the results, indicating that model differences for outweigh any observational issues for this analysis.

5/ ECS computed from abrupt 4xCO2 simulations is generally larger than when computed from abrupt 2xCO2 simulations. It is also an imperfect predictor of the warming predicted in transient simulations. It is not clear to me why the author considers ECS rather than TCR as a predictor of GHG or total historical warming.

It is imperfect but certainly useful, see figure 1b. The ECS values quoted here are derived from 4xCO2 simulations. While there are differences between the two types of experiments used to quantify ECS, the results should be highly correlated, and most ECS values are derived from 4xCO2 experiments that are part of the ubiquitous DECK family of experiments. We now treat the TCR in the same way as the ECS. Interestingly, our result for the ECS (3.4K) is larger than the IPCC best estimate (3K), while the TCR result (1.8 K) is the same as IPCC. Both are consistent with the IPCC uncertainty ranges.

6/ The author argues that his estimate of ECS is "very consistent with the AR6 estimate". Yet his estimate of the aerosol cooling (-0.19°C ± 0.14 °C) isn't consistent with the IPCC estimate (see bullet A.1.3 of the AR6 SPM that says "The likely range of total human-caused global surface temperature increase from 1850–1900 to 2010–2019 is 0.8°C to 1.3°C, with a best estimate of 1.07°C. It is likely that well-mixed GHGs contributed a warming of 1.0°C to 2.0°C, other human drivers (principally aerosols) contributed a cooling of 0.0°C to 0.8°C, natural drivers changed global surface temperature by –0.1°C to +0.1°C, and internal variability changed it by –0.2°C to +0.2°C"). Thus IPCC estimate is centered on -0.4°C rather than -0.2°C. It would be good to mention and discuss this apparent disagreement, otherwise there is little value in flagging the agreement on the ECS.

The adjusted NTCF-induced cooling is now just within the very wide 5-95% uncertainty range of the IPCC estimate (adjusted for different reference periods and accounting for ozone, 0.15-0.57 K) but indeed skews towards a smaller absolute best-estimate value than IPCC. I cannot fully explain this but note that excluding models that do not satisfy additivity considerably improves agreement amongst the remaining 7 models regarding the best-estimate cooling due to aerosols.

The smaller best-estimate aerosol-induced cooling suggested by the regression analysis is not entirely surprising. For individual models this has been noted, also for the group (by Gillett et al., 2021) but not quantified in quite the same way as I do here. Quite what physical shortcoming is causing this, I can only speculate.

7/ Bottom-up estimates of the aerosol radiative forcing (e.g. Bellouin et al., doi: 10.1029/2019RG000660, 2019) are also larger (i.e. more negative) than implied by the aerosol cooling inferred from this study. This does not invalidate the current analysis. Yet the fact that it is at odds with a number of other studies requires some discussion.

I now discuss Bellouin et al. They come up with upper and lower bounds of the aerosol-induced radiative forcing of -2 to -0.35 W m$^{-2}$, translating into roughly 0.18 to 1 K of cooling due to aerosols between 1850 and 2005-2015. Thus the reference periods are very similar to the one used here (1850-1899 to 2000-2014). Their PDF for the combined aerosol-radiation and aerosol-cloud effect peaks at about -0.7 W m$^{-2}$ or 0.35 K of cooling. My result is consistent with but skewing slightly smaller than Bellouin et al. I note that in the calculation, using only three models, where ozone is

explicitly included, I obtain 0.27+-0.08 K as the best estimate for the aerosol-induced cooling, close to the centre of the PDF of aerosol-induced cooling presented by Bellouin et al.

8/ Ocean diffusivity also affects the time evolution of the GSAT in models. How would biases in ocean diffusivity in the models affect the author's analysis?

I cannot say anything definitive about the role of ocean diffusivity. Models used here generally parameterize ocean eddies because of insufficient resolution. This may be a reason for general shortcomings in the models, e.g. misrepresentations of meridional heat flux in the ocean and consequent biases e.g. of high-latitude SST. However such process-oriented questions are far outside the scope of this paper and would require a process-oriented study to address.

9/ Lines 70-71, equations 4 and 5: the author appears to assume that alpha1 and alpha2 (beta1 and beta2) are independent variables so that their error variances can be summed but is this really the case?

These equations are now gone due to a change in method. Hence this comment is now moot. (I suppose they are actually not independent, so this wasn't quite the correct approach.)

10/ The reader needs to understand how DAMIP ensemble members for a given model were treated in the analysis.

The analysis only uses ensemble means. So I have formed, individually for every model, an ensemble mean of the available ensemble members for ts and tas (numbers are in table 1), and then merged these fields into the temperature field used in the rest of the analysis. This is explained in the text (around eq. 1).

Minor comments

Lines 19-20 is a bit of a truism.

Maybe but it remains correct. I have added that cumulative model improvements have not changed this situation.

Line 28: the decreasing aerosol loading is not a "feedback" but a forcing.

I have replaced "feedback" with "effect", i.e. how much warming this will produce.

Lines 33-34: sentence unclear, please reformulate.

I have reformulated the sentence. Hopefully it is now clear.

Figs 1 and 5: Plotting b versus a generally implies that b is on the y-axis. Doing the opposite is confusing.

I have reversed the axes.

Line 116: why four models and not all models?

This plot is now showing only 1 model (HadGEM3) to explain the concept. All other models are shown in figure A1. The aim was not to overload the figure.

Fig 2: is the plot for the model ensemble mean or for a particular member of DAMIP?

The plot is for the ensemble mean, as stated in the caption.

References

I am now discussing most of the below references, particularly Bellouin et al., Gillett et al., and Storelvmo et al..

Bellouin, N., J. Quaas, E. Gryspeerdt, S. Kinne, P. Stier, D. Watson-Parris, O. Boucher, K.S. Carslaw, M. Christensen, A.-L. Daniau, J.-L. Dufresne, G. Feingold, S. Fiedler, P. Forster, A. Gettelman, J. M. Haywood, F. Malavelle, U. Lohmann, T. Mauritsen, D.T. McCoy, G. Myhre, J. Mülmenstädt, D. Neubauer, A. Possner, M. Rugenstein, Y. Sato, M. Schulz, S. E. Schwartz, O. Sourdeval, T. Storelvmo, V. Toll, D. Winker, and B. Stevens, Bounding aerosol radiative forcing of climate change, Reviews of Geophysics, 58, e2019RG000660, doi: 10.1029/2019RG000660, 2019.

Charles, E., B. Meyssignac, and A. Ribes, 2020: Observational Constraint on Greenhouse Gas and Aerosol Contributions to Global Ocean Heat Content Changes. J. Climate, 33, 10579–10591, https://doi.org/10.1175/JCLI-D-19-0091.1.

Gillett, N.P., Kirchmeier-Young, M., Ribes, A. et al. Constraining human contributions to observed warming since the pre-industrial period. Nat. Clim. Chang. 11, 207–212 (2021). https://doi.org/10.1038/s41558-020-00965-9

Knutti, R. (2008), Why are climate models reproducing the observed global surface warming so well? Geophys. Res. Lett., 35, L18704, doi:10.1029/2008GL034932.

Storelvmo, T., Leirvik, T., Lohmann, U. et al. Disentangling greenhouse warming and aerosol cooling to reveal Earth's climate sensitivity. Nature Geosci 9, 286–289 (2016). https://doi.org/10.1038/ngeo2670

---

## Author Comment (AC2)

Christopher Smith: (The reviewer's comments are in red, responses in black.)

I thank Christopher Smith for some expert, balanced, constructive comments.

In this paper, Morgenstern uses 15 CMIP6 models contributing to the Detection and Attribution Model Intercomparison Project (DAMIP) to estimate the relative contributions to greenhouse gas and aerosol forcing in the present day, and uses the analysis to provide an emergent constraint on the equilibrium climate sensitivity (ECS). The two main findings of the paper are that (1) the ECS is in line with the IPCC headline assessment (likely range 2.5-4.0°C), a little lower than the models' unadjusted values, and (2) the aerosol warming contribution is only one-third as large as the models' unadjusted values and substantially less negative than the IPCC headline assessment of -0.5°C (-1.0°C to -0.2°C very likely range). If the second point is correct, the implications for future climate change are huge, in the sense that the masked warming by aerosols is small and climate would not be expected to warm by a large amount in any future emissions mitigation scenario.

The analysis hinges solely on CMIP6 models. Therefore, there is a risk that the conclusions are over-confident.

That is indeed a risk. I have now modified the discussion to make explicit that there are different possible explanations for the findings, and also provide a more in-depth discussion versus the IPCC assessment and other lines of evidence. This shrinks the discrepancy versus the IPCC estimates. I have also added as a very well-performing model MPI-ESM1-2-LR which simulates an aerosol-induced cooling of just 0.3 K, and requires relatively small adjustments to bring its historical temperature evolution into agreement with HadCRUT5.

One weakness of a purely CMIP6 historical approach is that the models are all forced with the same (largely uncertain) spatial and temporal aerosol emissions data set. It's possible that the era of strong aerosol cooling in many CMIP6 models (the whole 20th Century in some models, but we see that many see a bit of a step change around 1950) could be an artefact of the forcing dataset. It could also be because the models are producing overly sensitive responses to aerosols during these time periods, which may also be a factor of some models having high climate sensitivity. I don't believe anybody has put forward a convincing argument one way or the other yet, though Smith & Forster (2021) and Flynn et al. (2023) have both tried to answer this question.

I now discuss this situation and the references in more depth. I don't have a conclusive answer either as to whether this is a general model shortcoming or a property of the CMIP6 emissions dataset. The dominance of models requiring scaling-down of aerosol-induced cooling may indicate the former.

The point I'm trying to make is that if the "shape" of the aerosol cooling time series differs between the models and reality, then an optimal fingerprinting approach may try to mitigate the effect by selecting regression coefficients $\beta_1$ that are less than one. This will reduce any error in the total warming time series when the individual components are summed up, and also implies that $\alpha_1 < 1$ to balance out the positive effect of the GHG warming. We indeed observe that $\beta_1 < 1$ in all models and $\alpha_1 < 1$ in all but two models. The author "normalizes" this approach by also determining the regression coefficients compared to each model's historical run, which is a good idea. However, interestingly again, the regression coefficients $\alpha_2$ and $\beta_2$ are also usually less than one (described in lines 150-151 as a lack of additivity). This could be suggestive of the regression approach attempting to minimize residual errors caused by natural variability rather than a genuine lack of additivity, though it should be noted that the omission of ozone and land use forcings may not be insignificant.

To investigate this, perhaps a rolling mean filter applied to the $T_{h*}$ terms in eqs. (1) and (2) could be investigated. CanESM5 hints at this effect: at 25 ensemble members, its model-derived internal variability is small, and it is the only model where $\alpha_2$ and $\beta_2$ are both greater than (and are also quite close to) 1, and the noted HadGEM3's approximate linear behavior has a 60-member hist-aer ensemble to draw upon.

Small changes to the regression are now improving the situation. Also I now use the "historical" simulations to restrict the analysis to additive models. CanESM5 is never considered "additive". Towards the end of the period its simulated warming is larger than can be explained by aerosols and GHGs. I now discuss the role of ozone in some more detail. Land use changes are very small climate forcers, on the global scale (IPCC AR6).

The shape of the historical aerosol cooling is something that we investigated in Smith et al. (2021). If we allow this to vary more (taking CMIP6 as an ensemble of opportunity, fitting a non-linear functional form and sampling the parameters) then we can construct aerosol forcing histories that do permit strong cooling and are still consistent with observations.

Therefore, it is my working hypothesis that the author finds a weak contribution to historical aerosol cooling because the historical shape of aerosol cooling (and forcing) is a poor fit to that implied by global temperatures and not easily resolved using a linear combination of GHG and aerosol attributed warming, and not necessarily because the present-day level of cooling is incorrect in the models (though historically, it likely was in some).

This would undermine most attribution approaches that rely on linear regression. I note that now eight models satisfy additivity for both the GHG induced warming and the aerosol-induced cooling. For all these models the regression produces very close fits to the observed temperature.

I'm also curious about the slightly different estimate for the present-day aerosol cooling to Gillett et al. (2021), who also did an optimal fingerprinting approach with CMIP6 DAMIP models and found aerosol cooling to be -0.7 to -0.1 °C. In their fig. 2b, it can be seen that aerosol regression coefficients > 1 for some models, though typically they also are in the 0 to 1 range. It would be useful to compare the differences and methods between the two papers.

Gillett et al. include in their analysis all models without regard to additivity, but only with one ensemble member each. Their regression model is very similar to mine. Their figure 2b indicates pretty small regression values for some of the same models that feature in my analysis (ACCESS-ESM1-5, CESM2, GISS-E2-1-G, HadGEM3-GC31-LL) and values greater than 1 for MIROC6 that has smaller aerosol-induced cooling. So I think their analysis is consistent with me, factoring in that I use ensemble means which reduces single-model uncertainties.

I do not want to come over as overly critical. It is a thorough yet concise paper, mathematically rigorous but not over-complicated, and the figures, equations and structure are clear and logically organized. Given the sensitivity and importance of the topic, the results should be contextualized relative to the IPCC assessment, which used more lines of evidence than solely CMIP6 (analogous to the ECS).

Indeed. I now state several times how the results compare to the AR6 assessed values. I will leave it to AR7 to place these results into context with other lines of evidence (paleo climate, process-oriented studies, etc), a full discussion of which is out of bounds for this paper. I believe though that in studies such as Gillett et al. (2021), non-additivity means that several models have to be removed from the analysis, which can substantially affect the overall conclusions.

Minor comments:

Abstract line 2: suggest replacing "anomalously large" with simply "larger". I don't think ECS of 5.6 or 5.7 K can be categorically ruled out.

Indeed. IPCC calls simulations produced by these models "low-likelihood, high impact". I follow your suggestion.

Line 35: "heuristic regression". I might be showing my ignorance here but I don't know what this is. It seems to be defined as a machine learning concept (https://dl.acm.org/doi/abs/10.1145/503810.503823). Was this the method used? Eqs. (1) and (2) look more like regular least-squares.

No, the naming coincidence is accidental. I now drop the word "heuristic". Indeed the approach is an ordinary least-squares regression method.

Line 60: I wonder why 1920-2020 and not the whole time period. Are results sensitive to the start date? I imagine they'd be very different if you used 1970.

I have now changed to a "symmetrical" regression (by introducing intercepts) and have expanded the regression period to 1850-2020 (or 1850-2014, for four models). The results are qualitatively similar to using the later start date.

Line 72: "single variable uncertainties": would this be standard error?

Indeed this is the standard error.

Line 219-220: This statement of models exceeding 0.5K cooling being unrealistic is too strong.

I have rephrased the line to make it more accommodating of other possibilities and avoid categorical statements like this.

Line 246: "global warming": since we're also talking about aerosol cooling, I suggest being more general: "anthropogenic climate change".

The term "global warming" is used here deliberately because the paper is only concerned with surface temperature.

Sign convention for any time you talk about a cooling, e.g. lines 9, 218, 219: a minus cooling is a double negative.

Those "minus" signs are now generally removed.

References:

I now cite and use all of these references.

Flynn et al. (2023): https://acp.copernicus.org/articles/23/15121/2023/acp-23-15121-2023.html

Gillett et al. (2021): https://www.nature.com/articles/s41558-020-00965-9

Smith et al. (2020): https://acp.copernicus.org/articles/20/9591/2020/acp-20-9591-2020.html

Smith et al. (2021): https://agupubs.onlinelibrary.wiley.com/doi/full/10.1029/2020JD033622

Smith et al. (2021): https://agupubs.onlinelibrary.wiley.com/doi/full/10.1029/2020JD033622

---

## Author Response (AR2)

**Reply to reviewer 1:**

I thank the reviewer for this additional review and some very useful comments. Review comments are in red, my responses in black.

The author has modified his methodology following the reviewers' comments and as a result his study has become more rigorous and more convincing. There remains a couple of issues, and I recommend that the author addresses at least the first one before the manuscript be accepted.

First the author may have misunderstood my comment on the "amalgamation" of the surface temperature product. The problem is not so much using SST instead of SAT over the open ocean. After all, over the open ocean, the trend in SAT must be very close if not identical to the trend in SST. The problem is that some surface temperature products such as HadCRUT5 use SST over the open ocean but SAT over land and sea ice. As the fraction is sea ice has been decreasing over the years, this makes a small but documented difference. Hence Eq 1 on line 88 is not consistent with HadCRUT5 in that HadCRUT5 does not consider the land fraction but the land and sea ice fraction. This is described in the SI of the Morice et al (2021) JGR paper: "For the HadCRUT5 analysis the weighting is also dependent on sea ice coverage, with sea ice regions treated as land" and as result the f in their equation varies in time. The sentence about HadCRUT5 on line 85 is thus not correct.

I have corrected this issue. Now I perform the analysis according to Eq. 1 which is what the reviewer suggests. Along with an increase in ensemble sizes for some models due to a general trawl for new data that I have conducted for this revision, but also a decrease for some models with small ensemble sizes due to unavailability of the siconc/siconca variable, this change in the base data has caused some minor numerical changes of the results which are however not fundamental.

Second I am always a little dubious of the relevance of comparing a model ensemble mean of historical simulations (i.e. the average between many historical realizations) with the historical observations (i.e. only one possible realization), especially for models that show a large amount of low-frequency internal variability. Sampling (randomly) one member, as done in Gillett et al (2021) and in many emerging constraint studies, is probably no better. I am not sure what the best way of doing would be (some sort of a probabilistic approach?) but I think that this problem has plagued the community for too long. At least the author should recognize this issue.

I agree that this is an issue; it is now mentioned, but not comprehensively discussed, in the paper. A further development would be to turn the theory in this paper into a probabilistic approach as suggested. This is beyond this paper though. When I find time, I may write a sequel to this paper laying out this theory.

The historical aerosol cooling inferred by the author is on the low end of other estimates. This may well be true and this study is solid enough to be eventually published but I note that there continues to be conflicting evidence on the magnitude of the aerosol forcing (and its contribution to cooling) during the historical period. This recent study (doi: 10.1038/s43247-024-01324-8) attributes the recent increase in the Earth Energy Imbalance (EEI) to fading aerosols and it is hard to reconcile with a small aerosol effect.

Thanks for pointing out this paper. Results in this paper at least superficially appear inconsistent with the results presented here. I can only acknowledge this problem; further research is needed to resolve it.

Minor comments

Line 23: p.a. => per annum

This is now spelled out.

Line 90: "surface temperature" should have read "land surface temperature"

I use the whole "surface temperature" field (ts) and the whole surface-air field (tas) but the weighting is such that over land, away from the coast, I actually use "surface-air temperature" not "land surface temperature". I think the present formulation, with the corrected Eq. 1, is now clear.